# Prediction of the Presence of Targetable Molecular Alteration(s) with Clinico-Metabolic ^18^ F-FDG PET Radiomics in Non-Asian Lung Adenocarcinoma Patients

**DOI:** 10.3390/diagnostics12102448

**Published:** 2022-10-10

**Authors:** Nicolas Aide, Kathleen Weyts, Charline Lasnon

**Affiliations:** 1UNICAEN, INSERM 1086 ANTICIPE, Normandy University, 14000 Caen, France; 2Nuclear Medicine Department, Comprehensive Cancer Centre F. Baclesse, UNICANCER, 14000 Caen, France

**Keywords:** adenocarcinoma, FDG, lung cancer, molecular analysis, pet, radiomic

## Abstract

This study aimed to investigate if combining clinical characteristics with pre-therapeutic ^18^ F-fluorodeoxyglucose (^18^ F-FDG) positron emission tomography (PET) radiomics could predict the presence of molecular alteration(s) in key molecular targets in lung adenocarcinoma. This non-interventional monocentric study included patients with newly diagnosed lung adenocarcinoma referred for baseline PET who had tumour molecular analyses. The data were randomly split into training and test datasets. LASSO regression with 100-fold cross-validation was performed, including sex, age, smoking history, AJCC cancer stage and 31 PET variables. In total, 109 patients were analysed, and it was found that 63 (57.8%) patients had at least one molecular alteration. Using the training dataset (*n* = 87), the model included 10 variables, namely age, sex, smoking history, AJCC stage, excessKustosis__HISTO_, sphericity__SHAPE_, variance__GLCM_, correlation__GLCM_, LZE__GLZLM_, and GLNU__GLZLM_. The ROC analysis for molecular alteration prediction using this model found an AUC equal to 0.866 (*p* < 0.0001). A cut-off value set to 0.48 led to a sensitivity of 90.6% and a positive likelihood ratio (LR+) value equal to 2.4. After application of this cut-off value in the unseen test dataset of patients (*n* = 22), the test presented a sensitivity equal to 90.0% and an LR+ value of 1.35. A clinico-metabolic ^18^ F-FDG PET phenotype allows the detection of key molecular target alterations with high sensitivity and negative predictive value. Hence, it opens the way to the selection of patients for molecular analysis.

## 1. Introduction

Lung cancer is the leading cause of cancer death in France as well as the world [1]. It is also one of the main worldwide indications of ^18^ F-fluorodeoxyglucose (^18^ F-FDG) positron emission tomography (PET) in nuclear medicine departments [2] and a research topic of major interest. Recently, many publications have explored the association of tumour mutations, especially epidermal growth factor receptor (EGFR), anaplastic lymphoma kinase (ALK), Kirsten rat sarcoma viral oncogene homolog (KRAS), and ROS1 mutations, with ^18^ F-FDG radiomics features for in vivo non-invasive diagnostic or prognostic purposes [3,4,5].

Meanwhile, in oncology genomics, next-generation sequencing (NGS) that allows the detection of multiple anomalies on different genomics scales, especially mutations and structural variations, supplanted traditional methods based on multiple tests. The strength of NGS is its ability to be performed with a small amount of tissue from a single biopsy [6], a single extraction, and a single test, which is both time- and cost-effective [7,8,9]. Although the whole genome sequencing remains a gold standard in oncology genomics, large DNA-based NGS selected panels conducted on cancer tissue samples of patients (and maybe in the near future on circulating cell-free DNA) have arisen as convenient alternatives [10]. 

These panels aim at identifying targetable molecular alterations helpful for the personalised treatment of patients. In non-small cell lung cancer (NSCLC), and specifically in lung adenocarcinoma, the NGS is increasingly carried out at diagnosis, since the first line of treatment depends on multiple molecular targets [11], but no consensus has been reached until now. The current hot topic is to determine who will benefit from NGS panels and when in the care timeline. 

In this regard, the present study aimed to investigate the combination of usual clinical characteristics (i.e., sex, age, smoking history, cancer stage according to the American Joint Committee on Cancer (AJCC)) and pre-therapeutic ^18^ F-FDG PET radiomics. The purpose was to find out if it could predict the existence of molecular alterations in key molecular targets in lung adenocarcinoma (namely EGFR, v-raf murine sarcoma viral oncogene homolog B1 (BRAF), KRAS, neuroblastoma RAS viral oncogene homolog (NRAS), mesenchymal–epithelial transition (MET), serine/threonine kinase 11 (STK11), phosphatidylinositol 3-kinase catalytic alpha polypeptide (PIK3CA), ALK and ROS1) to screen patients who are more likely to benefit from molecular testing. 

## 2. Materials and Methods

### 2.1. Population

This non-interventional monocentric study had a retrospective design. The study population included all patients with newly diagnosed lung adenocarcinoma who had an NGS exploration of panel Colon and Lung Cancer Panel v3 (CLv3). The subjects were selected from those patients who referred to our PET unit for their initial staging between January 2018 and December 2019. Their sex, age, smoking history, and AJCC stage were recorded. The institutional review board approved the study and waived the need for informed signed consent. Following the European General Data Protection Regulation, data collection for this study was approved by the national committee for data privacy, with the registration code of N° F20210309115801.

### 2.2. Immunohistochemistry

Automated immunohistochemistry using a Ventana Bench Mark Ultra was performed on 4 μm-thick paraffin sections of biopsy with clone D4D6 for ROS1 (pre-diluted). A complementary fluorescence in situ hybridisation (FISH) was systematically performed in case of positivity [12].

### 2.3. Next-Generation Sequencing Panel Colon and Lung Cancer Panel v3 Analysis [13]

Analyses were carried out using a fixed and included paraffin sample. The tumour genomic DNA extraction was conducted with the Q1Aamp DNA FFPE Tissue Kit on Q1Acube (Q1AGEN). The NGS was performed using an Ion Personal Genome Machine (Life Technologies, Villebon sur Yvette, France). The average depth was >500X; on target >90%. Bioinformatic analyses (alignment and call of variants and annotations):−Life Technologies: Torrent Suite (version 5.6), Variant Caller (version 5.6), Ion Reporter (version 5.6)−Nextgene (version 2.4.1.2, Softgenetics, State College, PA, USA).

The copy number variant analysis was expressed as the ratio of mean depths to amplicons ± 2. The detection limit was set to 3% for punctual mutation and 5% for insertion/deletion for a minimum depth of 100X per amplicon. Variations of sequences recognised as non-pathogenic were not mentioned. 

### 2.4. Positron Emission Tomography Acquisition and Analysis

Patient care and ^18^ F-FDG administration (3.0 MBq/kg) were performed according to the guidelines of the European Association of Nuclear Medicine for oncologic examinations [14]. The PET acquisitions were acquired on 2 systems: (1)A TrueV analogic PET/CT (Siemens Healthineers, Erlangen, Germany) with three iterations and 21 subsets with point spread function (PSF) reconstruction (2.0 × 4.0 × 4.0 mm^3^ voxels). The PET emission acquisition was performed from skull to mid-thighs for 2 min and 40 s and 3 min and 40 s per bed position for normal-weight and overweight patients, respectively.(2)A Vereos digital PET/CT (Philips Medical Solutions, USA) with two iterations and 10 subsets with PSF reconstruction (2 mm^3^ voxels). The PET emission acquisition was performed from the skull to mid-thighs for 2 min per bed position regardless of the body habitus of the patients.

An experienced nuclear physician drew volumes of interest over the hypermetabolism of the primary lung lesion using a gradient-based delineation, showing to outperform threshold-based methods in terms of accuracy and robustness [15] (PET edge) on MIM software (version 5.6.5, MIM Software Inc., Cleveland, OH, USA) and recorded as RTstruct files. Afterward, the RTstruct files were uploaded in LifeX 6.3 software [16], and the automatic close function (3D dilatation followed by erosion of 10 voxels) was systematically applied to account for any hypo-metabolic area(s), such as necrotic parts of the tumour volume. No other changes were made, especially freehand modifications. A small lesion size leading to a volume of interest containing less than 64 voxels was an exclusion criterion according the LifeX software procedural standards. Indeed, this is a well-known impairment for the calculation of metabolic 18 F-FDG-PET heterogeneity [17].

The following parameters, fulfilling the Image Biomarker Standardization Initiative [18], were extracted from PET images using an absolute resampling of 64 grey levels and standardised uptake value (SUV) comprised between 0 and 30:−Conventional parameters: SUV_mean_, SUV_max_, metabolic tumour volume, and total lesion glycolysis−Histogram parameters: skewness__HISTO_, kurtosis__HISTO_, excessKustosis__HISTO_ entropy_log2__HISTO_, and uniformity__HISTO_−Shape parameters: sphericity__SHAPE_ and compacity__SHAPE_−Grey-Level Co-Occurrence Matrix (GLCM) parameters: inverse difference__GLCM_, angular second moment__GLCM_, variance__GLCM_, correlation__GLCM_, joint entropy__GLCM_, and dissimilarity__GLCM_−Neighbouring grey-level dependence matrix (NGLDM) parameters: coarseness__NGLDM_, contrast__NGLDM_, and busyness__NGLDM_−Grey-level zone length matrix (GLZLM) parameters: SZE__GLZLM_, LZE__GLZLM_, LGZE__GLZLM_, HGZE__GLZLM_, SZLGE__GLZLM_, SZHGE__GLZLM_, LZLGE__GLZLM_, LZHGE__GLZLM_, GLNU__GLZLM_, ZLNU__GLZLM_, and ZP__GLZLM_

Detailed specifications for these parameters are available on the Image Biomarker Standards Initiative website [19].

A post-reconstruction harmonisation was run using ComBaT for all PET parameters found to be statistically significantly different between PET systems [20]. 

### 2.5. Statistical Analysis

The data are presented as mean values unless otherwise specified. The PET parameters extracted from the 2 PET systems described above were compared using a Mann–Whitney non-parametric test before and after ComBat harmonisation applying Bonferroni correction. A process was used by randomly splitting the data in training (87 patients, 80%) and test (22 patients, 20%) datasets. Instead of creating a separate validation dataset, a k-fold cross-validation was used with the training dataset to tune model parameters.

To select features of interest and build a model of prediction, the LASSO regression with a binary response variable (0: no molecular alteration, 1: at least one molecular alteration) was performed on the training dataset of patients with 100-fold cross-validation, including the following explanatory variables: sex, age, smoking history, AJCC stage and all the previously described PET variables. The optimal pred.core cut-off value was determined using the receiver operating characteristic curve. This pred.score cut-off value was then tested on the unseen test dataset. Sensitivity (also known as recall), specificity, positive predictive value (also known as precision), negative predictive value, positive likelihood ratio (LR+), negative likelihood ratio (LR−), and accuracy were reported. LRs are defined as follows:(1)$$\mathrm{LR}+=\frac{sensitivity}{1-specificity}$$
(2)$$\mathrm{LR}-=\frac{1-sensititivy}{specificity}$$

The LR includes the sensitivity and specificity of a test into a single measure. The best diagnostic test to detect the disease is the one with the larger LR+ [21,22].

Statistical analysis and graphs were made using XLSTAT software (version 2019: Data Analysis and Statistical Solution for Microsoft Excel. Addinsoft). A *p*-value of less than 0.05 was considered statistically significant unless otherwise specified.

## 3. Results

### 3.1. Patients and Next-Generation Sequencing Characteristics of the Entire Data Population

In total, 111 patients were screened from January 2018 to December 2019. Two patients were excluded due to the small volume of their lesions (3.0 and 1.4 mL). Finally, 109 patients were included. The database was composed of 34 female and 75 male subjects (sex ratio M/F = 2.2) with a median age of 66 years (range: 37–86). Moreover, 96 patients (88.1%) had a history of smoking. There were 19 (17.4%), 37 (34.0%), and 53 (48.6%) patients at AJCC stages of I or II, III, and IV, respectively.

In addition, 63 patients (57.8%) had at least one molecular alteration. Among them, 53, 8, and 2 patients had one (84.1%), two (12.7%), and three alterations (3.2%), respectively. Figure 1 displays a detailed description. With regard to EGFR mutations, nine, three, and two were in exon19, exon20, and exon21, respectively. The KRAS mutations were all in the exon2 except for one patient. It is noteworthy that no HER2 mutation occurred. 

The median delay between the completion of the biopsy and the availability of the results for all previously described mutations was 23 days. For 100 patients (91.7%), the PET/CT examination was performed before the availability of the genetic results with a median interval of 39 days. For 69 patients (63.3%), the PET/CT was even performed before the biopsy with a median interval of 21 days. Table 1 summarises the clinical characteristics of the training and test datasets of patients. 

### 3.2. Positron Emission Tomography Data Harmonisation

In total, 51 patients (46.8%) underwent their examinations on the TrueV analogic system and 58 (53.2%) underwent their examinations on the Vereos digital system. None of the conventional and histogram PET parameters were significantly different between PET systems (Appendix A). The parameters found to be statistically different among PET systems were compacity__SHAPE_, variance__GLCM_, correlation__GLCM_, dissimilarity__GLCM_, coarseness__NGLDM_, contrast__NGLDM_, SZE__GLZLM_, LZE__GLZLM_, LGZE__GLZLM_, SZLGE__GLZLM_, LZHE__GLZLM_, and ZP__GLZLM_. However, after ComBat harmonisation, only LZHE__GLZLM_ remained different among PET systems (*p* < 0.0001, Table 2); therefore, this parameter was not further considered.

### 3.3. Construction of Prediction Model Using a Lasso Regression with a Cross-Validation on the Training Dataset (n = 87)

The model included 10 variables, namely age, sex, smoking history, AJCC stage, excessKurtosis__HISTO_, sphericity__SHAPE_, variance__GLCM_, correlation__GLCM_, LZE__GLZLM,_ and GLNU__GLZLM_ (Figure 2 and Figure 3). The optimal lambda determined by cross-validation was equal to 0.030, and the corresponding coefficients can be seen in Appendix A. 

### 3.4. Relationship between Variables Included in the Lasso Regression Model

Significant correlations were observed among PET variables of interest, which are summarised in detail in Table 3. The strongest correlation was observed between variance__GLCM_ and LZE__GLZLM_ with ρ = −0.702 and *p* < 0.0001. There was no correlation between age and PET variables of interest (Table 3). Moreover, there were no differences in PET variables of interest in terms of sex, smoking history, and AJCC stages (Table 4). 

### 3.5. Comparison of Variables Included in the Lasso Regression Model between Patients with and without Molecular Alteration(s)

There was no significant difference between patients without any molecular alterations and patients with at least one molecular alteration in terms of age, excessKustosis__HISTO_, sphericity__SHAPE_, variance__GLCM_, correlation__GLCM_, LZE__GLZLM_, and GLNU__GLZLM_ (Figure 3). There was no association between molecular status and AJCC stage (*p* = 0.102, Figure 4). 

Sex and smoking history were significantly associated with the molecular status. More molecular alterations were observed in females (84.0%) compared to males (51.6%) (*p* = 0.005, Figure 3). The KRAS and EGFR mutations represented the great majority of molecular alterations in females, as 56.0% and 28.0% of them had KRAS and EGFR mutations, respectively. In addition, the only positive ROS1 translocation-related protein expression was observed in a woman. A complementary FISH was performed confirming the presence of an ROS1 translocation in 80% of the tumour cells visualised. In addition, in the training dataset, all non-smokers (*n* = 8) presented at least one molecular alteration (*p* = 0.017).

### 3.6. LASSO Regression Model Diagnostic Performances for Molecular Alteration(s) Detection in the Training Dataset (n = 87)

The receiver operating characteristic analysis for molecular alteration prediction using this LASSO regression model in the training dataset of patients found an area under the curve equal to 0.866 (95%CI = 0.792–0.941, *p* < 0.0001). The pred.score cut-off value (or operating point) was chosen by optimising utility with the aim of optimising the sensitivity of the test to lower the rate of false-negative cases (FN) as appropriate for a screening test. A cut-off value set to 0.48 led to a sensitivity of 90.6%, a negative predictive value of 80.8%, and an LR+ of 2.37 (specificity = 61.8%, positive predictive value = 78.7%, accuracy of 79.3%). It should be mentioned that in this configuration, 26 patients (29.9%) tested negative. Among them, only five patients had FN results, and none of them had EGFR mutations (one ALK/KRAS, one KRAS, two STK11, and one PIK3CA). Among the 61 patients tested positive, 49 (80.3%) were true positives. These true-positive patients had a median age of 69 years (range = 38–86), 56.3% were males, and 83.3% had smoking history.

### 3.7. Prediction Model Screening Performances on the Unseen Test Dataset (N = 22)

Applying this cut-off value in the validation dataset of patients, the test presented a sensitivity of 90.0%, a negative predictive value of 80.0%, and an LR+ equal to 1.35 with one FN result among the five negative tests (22.7%) concerning a KRAS mutation. There were nine true-positive patients who had a median age of 66 years (range = 55–83), 22.2% were males, and 77.8% had smoking history. Representative images of a true-positive patient are shown on Figure 5.

## 4. Discussion

Usage of a model based on clinical and PET radiomics appears to be a promising strategy for screening at the time of diagnosis of lung adenocarcinoma patients who may have a targetable molecular alteration. This could certainly reduce unnecessary costs by avoiding the need to test patients for whom we could know that there is little chance of finding molecular conditions. Moreover, it helps speed the management of these patients by eliminating the need to wait for the results of an unnecessary test. For instance, in the present study, PET/CT was usually performed more than one month before the availability of genetic analyses results (median = 39 days). 

At our institution, somatic mutation detection using immunochemistry and NGS Panel CLv3 and single tests for ROS1 and ALK mutations was performed for nearly all lung adenocarcinoma patients at the time of diagnosis, and finally, almost half of them had negative findings (42%). It is worth noting that frequencies of molecular alterations observed in our database are representative of those previously observed in a Western population with lung cancer [23]. Therefore, it can be said that using the proposed regression model would have avoided 31 molecular tests (26 in the training and five in the test datasets). In other words, applying this strategy could spare almost 1/3 of molecular testing at the cost of six false-negative tests (5.5%).

The included model was a mix of clinical characteristics (age, sex, smoking history, and AJCC stage) and PET characteristics (excessKustosis__HISTO_, sphericity__SHAPE_, variance__GLCM_, correlation__GLCM_ LZE__GLZLM_, and GLNU__GLZLM_). In the sight of the LASSO regression coefficients, the strongest predictive variables were tumour correlation__GLCM_, sphericity__SHAPE_, sex, and smoking history. Sex and smoking history were significantly different between patients with and without molecular alterations on univariable analysis with an increased risk of molecular alterations in females and non-smokers. This is somewhat concordant with the results of previous studies indicating the higher rate of EGFR mutations [24,25,26], KRAS mutations [27,28], ROS1 gene fusion [29,30], and STK11 [31] expression in females. Moreover, non-smokers with lung cancer (mostly adenocarcinomas) have more tumour genetic mutations than smokers, according to the results of the sequencing-based studies [32,33,34]. To date, to predict whether a lung adenocarcinoma might harbour targetable genetic alterations, the clinician considered the aforementioned factors related to the pathobiology of smoke-associated tumours. Thus, it is interesting to note that in our patient dataset, many of the patients with one or more molecular alterations tested positive by the model were male and had a history of smoking. For example, in our training dataset, the true-positive patients were males in 56.3% of cases and had a history of smoking in 83.3% of cases. This supports the fact that the ^18^ F-FDG/PET signature may have additional value over the already common clinical stratification criteria.

Correlation__GLCM_ represents the linear dependency of grey levels in GLCM and sphericity is a measure of the roundness of the shape of the tumour region relative to the sphere. It should be noted that correlation__GLCM_ was among the parameters that needed to be harmonised between the two PET systems. This is in line with the results of previous studies demonstrating the impact of reconstruction parameters on conventional PET metrics and texture features in NSCLC [2,35] and hence, the need for harmonising standards. 

Moreover, when it comes to the comparison of analogic and digital PET quantitative variables analysis, it is important to point out that conventional and histogram parameters were not found to be different from our systems in this reconstruction configuration. A harmonisation process was needed only for some second- and third-order textural features and only one failure of the process was observed with LZHGE__GLZLM_. 

These findings demonstrated for the first time that textural features extracted from digital and analogic PET systems can be pooled using harmonisation strategies currently under development [20,36]. However, it has been shown that applying a smoothing filter with a large kernel as per EARL procedure [37] or using a larger voxel size can lead to the loss of accuracy of radiomics metrics for tumour characterisation purposes [38]. 

The study had some limitations; first, these encouraging results need to be confirmed by a larger multi-centre study. Furthermore, their extrapolation to other populations, for which the repartition of histological subtypes and mutational status could be different, needs to be investigated [39,40,41]. Moreover, to ensure its translation into clinical practice, a worldwide harmonisation strategy is needed, and the development of dedicated software for automatic computation of the model equation seems mandatory. Given the flourishing number of models of this type in the oncology literature for a multitude of hypotheses, this axis of development should be carried out in the short term. 

Secondly, since the great emulation in the framework of precision oncology and genetics, our knowledge will surely be in constant evolution, and a strategy that works today will have to be constantly adapted according to new discoveries in the field. Thirdly, the molecular analysis was usually performed on biopsies with the risk of spatial tumour heterogeneity and missing some tumour molecular alterations. For example, Swanton et al. noted that many oncogenic alterations were only identified in specific tumour locations generating tumour heterogeneity [42]. Similarly, Pelosi et al. micro-dissected several tumour regions of different architectures from 20 adenocarcinomas and revealed that 60% of these tumours had intra-tumour molecular heterogeneity [43]. Therefore, it is possible to wonder if some tests considered as false-positive results in our study were not linked to a biopsy sampling error. Indeed, metabolic tumour characterisation considers the entire tumour volume and not just a sample. This can be a strength together with its non-invasiveness. 

Finally, spatiotemporal heterogeneity could also be considered. In this study, the strategy was explored at diagnosis, but another time point during patient management could be investigated, as no consensus has yet been reached [44]. The few patients with false-negative tests at diagnosis might benefit from a molecular analysis at another time of their treatment.

In conclusion, screening non-Asian lung adenocarcinoma patients at the time of diagnosis by means of a model including clinical parameters and ^18^ F-FDG/PET radiomics before performing a tumour molecular analysis seems to be an efficient strategy. It allows predicting the existence of key molecular target(s) with high sensitivity.

## Figures and Tables

**Figure 1 diagnostics-12-02448-f001:**
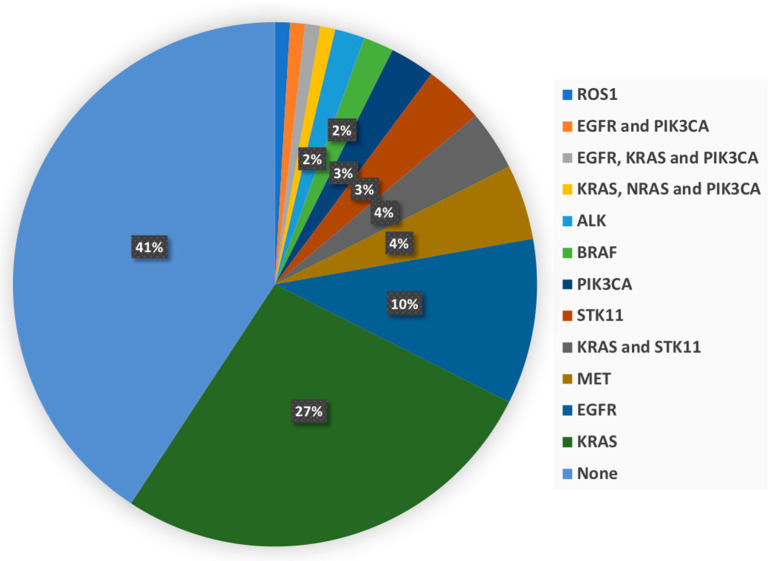
Molecular alterations description of patients.

**Figure 2 diagnostics-12-02448-f002:**
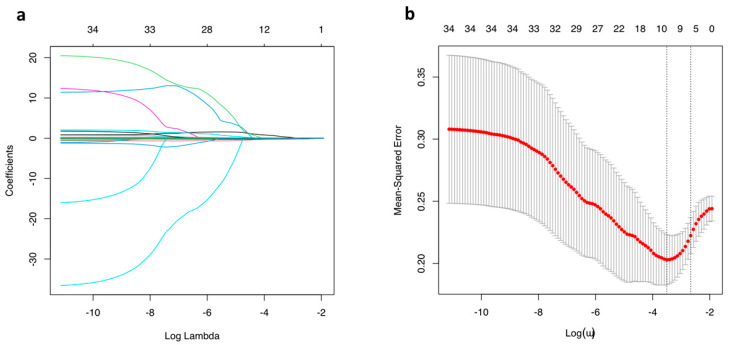
Coefficients (**a**) and cross-validation (**b**) plots of the LASSO analysis.

**Figure 3 diagnostics-12-02448-f003:**
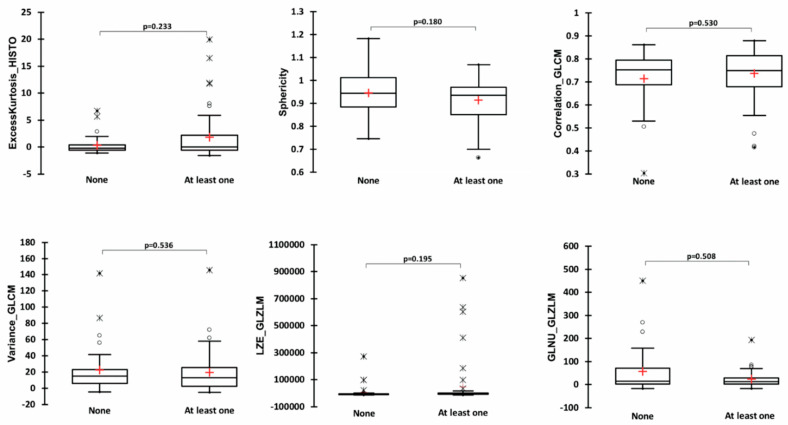
Comparison of patients without any molecular alteration and patients with at least one in terms of all positron emission tomography quantitative variables included in the model.

**Figure 4 diagnostics-12-02448-f004:**
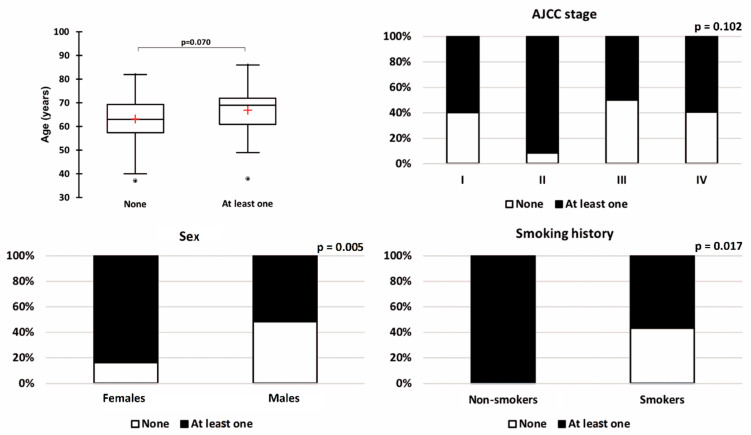
Comparison of patients without any molecular alteration and patients with at least one for all the clinical variables included in the model.

**Figure 5 diagnostics-12-02448-f005:**
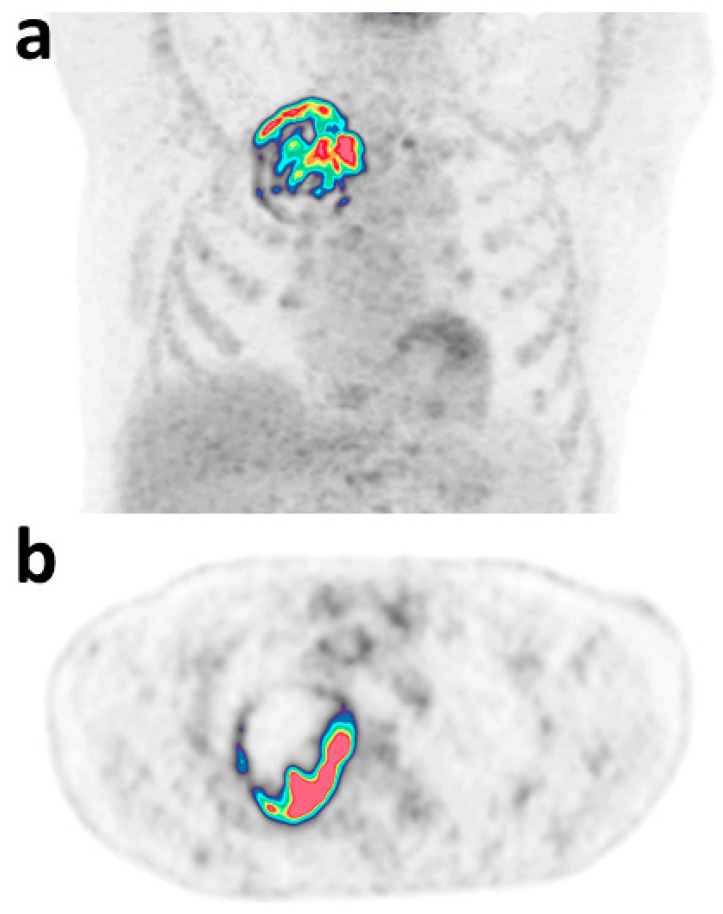
Representative maximum intensity projection (**a**) and axial (**b**) images of true-positive patients with MET gene alteration and a pred.score equal to 0.92. Model variables were: age = 70 years, sex = male, smoking history = yes, AJCC stage: IIIA, excessKurtosis__HISTO_ = 11.72, sphericity__SHAPE_ = 0.83, variance__GLCM_ = 24.40, correlation__GLCM_ = 0.87, LZE__GLZLM_ = 9188.94, GLNU__GLZLM_ = 76.81.

**Table 1 diagnostics-12-02448-t001:** Molecular and clinical characteristics of patients.

		Training Group (*n* = 87)		Test Group (*n* = 22)	
Variable	Categories	Frequencies	%	Frequencies	%
**Molecular alterations**	None	34	39.1	12	54.5
	At least one	53	60.9	10	45.5
**Gender**	Female	25	28.7	9	40.9
	Male	62	71.3	13	59.1
**Smoking history**	No	8	9.2	5	22.7
	Yes	79	90.8	17	77.3
**AJCC stage**	I or II	17	19.5	2	9.1
	III	28	32.2	9	40.9
	IV	42	48.3	11	50.0

**Table 2 diagnostics-12-02448-t002:** Comparison of TrueV and Vereos positron emission tomography quantitative variables before and after the ComBat harmonisation process. The only variables found to be significantly different between the TrueV and the Vereos systems before the ComBat harmonisation process are presented here. To see the data for all variables, please refer to the exhaustive Appendix A (* according to Bonferroni correction, a *p*-value < 0.0016 was considered significant).

Variable	Min	Max	Mean	*p* Value *	Min	Max	Mean	*p* Value *
Shape parameters								
Compacity | TrueV	0.735	8.699	2.401		0.492	7.707	3.058	
Compacity | Vereos	1.311	11.225	4.439	**<0.0001**	1.027	7.727	3.048	0.963
GLCM parameters								
Variance | TrueV	2.169	207.117	28.928		0.383	145.670	19.352	
Variance | Vereos	1.019	60.837	10.932	**<0.0001**	−4.923	141.562	19.352	0.854
Correlation | TrueV	0.211	0.789	0.651		0.278	0.861	0.721	
Correlation | Vereos	0.356	0.945	0.784	**<0.0001**	0.305	0.879	0.721	0.892
Dissimilarity | TrueV	0.952	10.475	3.422		0.839	8.457	2.815	
Dissimilarity | Vereos	0.715	6411	2.281	**0.0002**	0.685	8.429	2.815	0.839
NGLDM parameters								
Coarseness | TrueV	0.001	0.070	0.018		−0.001	0.057	0.014	
Coarseness | Vereos	0.000	0.054	0.009	**0.001**	0.002	0.070	0.014	0.674
Contrast | TrueV	0.014	0.878	0.198		0.000	0.664	0.141	
Contrast | Vereos	0.008	0.618	0.091	**<0.0001**	0.014	0.942	0.141	0.774
GLZLM parameters								
SZE | TrueV	0.229	0.775	0.536		0.211	0.723	0.499	
SZE | Vereos	0.230	0.704	0.466	**0.0004**	0.252	0.748	0.499	0.802
LZE | TrueV	4.368	18,386.635	1127.261		−13,631.716	635,105.336	25,996.822	
LZE | Vereos	3.600	1,191,735.375	47,864.883	**<0.0001**	−8620.105	853,331.114	25,996.822	0.0002
LGZE | TrueV	0.004	0.225	0.034		0.002	0.166	0.024	
LGZE | Vereos	0.001	0.087	0.016	**<0.0001**	−0.001	0.148	0.024	0.631
SZLGE | TrueV	0.003	0.103	0.013		0.002	0.076	0.010	
SZLGE | Vereos	0.001	0.045	0.007	**<0.0001**	−0.001	0.080	0.010	0.353
LZHGE | TrueV	666.714	1,258,030.400	50,296.736		−155,560.617	43,339,050.699	156,236.634	
LZHGE | Vereos	1793.265	55,736,935.663	2,889,821.717	**<0.0001**	−527,621.307	39,784,583.618	1,561,236.634	**<0.0001**
ZP | TrueV	0.033	0.616	0.255		−0.014	0.537	0.196	
ZP | Vereos	0.007	0.625	0.144	**<0.0001**	0.054	0.695	0.196	0.839

**Table 3 diagnostics-12-02448-t003:** Spearman correlations matrix for positron emission tomography variables of interest and age. Values in bold are those that correspond to statistically significant results.

Variables	Age	ExcessKurtosis_ _HISTO_	Sphericity__SHAPE_	Variance__GLCM_	Correlation__GLCM_	LZE_ _GLZLM_	GLNU__GLZLM_
**Age**	**1**	−0.176	0.059	−0.105	0.067	0.169	0.083
**ExcessKurtosis_ _HISTO_**	−0.176	**1**	**−0.344**	**−0.369**	−0.075	**0.401**	**−0.232**
**Sphericity__SHAPE_**	0.059	**−0.344**	**1**	0.212	**−0.318**	−0.307	**−0.367**
**Variance__GLCM_**	−0.105	**−0.369**	0.212	**1**	−0.153	**−0.702**	0.167
**Correlation__GLCM_**	0.067	−0.075	**−0.318**	−0.153	**1**	**0.362**	**0.574**
**LZE_ _GLZLM_**	0.169	**0.401**	−0.307	**−0.702**	**0.362**	**1**	**0.263**
**GLNU__GLZLM_**	0.083	**−0.232**	**−0.367**	0.167	**0.574**	**0.263**	**1**

**Table 4 diagnostics-12-02448-t004:** Comparisons of positron emission tomography variables of interest according to sex, smoking history, and stage according to American Joint Committee on Cancer.

Variables		ExcessKurtosis_ _HISTO_	Sphericity__SHAPE_	Variance__GLCM_	Correlation__GLCM_	LZE_ _GLZLM_	GLNU__GLZLM_
**Sex**, mean (SD)	Females (n = 25)	1.849 (4.830)	0.904 (0.096)	20.010 (30.118)	0.743 (0.104)	35343.472 (133495.181)	19.109 (23.159)
	Males (n = 62)	0.967 (3.213)	0.936 (0.094)	20.979 (25.708)	0.722 (0.113)	30955.319 (145426.809)	44.254 (80.101)
	*p* value	0.685	0.140	0.611	0.453	0.476	0.748
**Smoking history,** mean (SD)	No (n = 8)	−0.270 (0.653)	0.930 (0.078)	19.039 (15.733)	0.744 (0.083)	−4966.920 (9276.960)	26.023 (27.943)
	Yes (n = 79)	1.372 (3.891)	0.927 (0.097)	20.869 (27.814)	0.726 (0.113)	35981.670 (147862.953)	38.143 (72.412)
	*p* value	0.260	0.891	0.670	0.903	0.608	0.812
**AJCC stage,** mean (SD)	I (n = 5)	4.180 (7.410)	0.969 (0.090)	14.265 (22.681)	0.574 (0.151)	−6434.922 (2239.339)	2.494 (1.497)
	II (n = 12)	1.271 (3.842)	0.886 (0.083)	15.924 (15.299)	0.755 (0.095)	−6371.951 (8811.401)	44.015 (57.732)
	III (n = 28)	0.426 (2.868)	0.950 (0.084)	21.956 (30.266)	0.736 (0.120)	29371.230 (119,758.438)	41.815 (62.151)
	IV (n = 42)	1.384 (3.619)	0.918 (0.102)	21.995 (27.999)	0.733 (0.091)	49,739.528 (17,7486.562)	35.952 (80.746)
	*p* value	0.092	0.186	0.862	0.066	0.329	0.195

## Data Availability

All data generated and analysed during this study are included in this published article. Raw data supporting the findings of this study are available from the corresponding author on reasonable request.

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
