# Peer review of "Prediction of the Presence of Targetable Molecular Alteration(s) with Clinico-Metabolic 18 F-FDG PET Radiomics in Non-Asian Lung Adenocarcinoma Patients"

_diagnostics, 2022, doi:10.3390/diagnostics12102448_

Round 1
Reviewer 1 Report
An outstanding manuscript, carefully written, on a subject of great importance for precision oncology.
A verification/standardization of the way of writing some of the references is necessary.
Reviewer 2 Report
This is a study which correlates some radiological parameters to the presence or absence of targetable molecular alterations in lung adenocarcinomas. The goal is to provide a framework to stratify patients that could benefit from molecular analysis in the clinical setting.
To date, to predict whether a given lung adenocarcinoma could harbor targetable genetic alterations, more frequent in non-smoke associated tumors, the clinician takes into account factors related to the pathobiology of smoke-associated tumors, namely gender, age, and of course smoking status. This study discusses this aspect in the discussion only briefly. If all your patients with the described radiological characteristics are all females/non-smokers, your data essentially does not add much to the currently described literature: no need for such radiological investigations in the clinical settings, gender/age/smoke are readily available from the first clinical assessment. In this view, the study is of little interest to oncology specialists, and only offers a descriptive radiological assessment of lung adenocarcinomas with targetable molecular alterations. Claims made in the discussion about the applicability of such radiological investigations in the clinical settings should be reviewed in this light, as this would be only a radiological descriptive study. I believe the Authors should focus on a specific aspect: what does your analysis adds to the already common stratifying criteria? What is the prevalence of patients with molecular targetable analysis showing your radiological signature that are males and smokers (if there are any)? Those patients would be the main finding of your study, in my opinion, and the manuscript should focus on them (or at least, address the issue).
The research, however, is conducted correctly, and apart from some issues regarding the English language, the manuscript is of quality.
Minor Points.
Page 2.
Line 15, specify what LR+ means.
Line 16, define NPV.
Page 3.
Line 6: reactive oxygen species (ROS) mutations? I think you mean the ROS1 gene... Nothing to do with reactive oxygen species.
Line 13 to 15: this claim needs to be backed by some reference.
Page 4.
Paragraph 2.1: provide some reference about the panel Colon and Lung Cancer Panel v3 (CLv3).
Paragraph 2.2: in the paragraph title, I think you mean "ImmunoHISTOchemistry".
Also, both antibodies used (D4D6 for ROS1 and 5A4 for ALK) are not companion diagnostics for detecting the underlying genetic alteration. This is a weak point of the manuscript.
2.4: many terms are somewhat obscure to a non-radiologist reader, like "skewness_HISTO, kurtosis_HISTO, excessKustosis_HISTO Entrop_log2_HISTO, and Uniformity_HISTO" as in the abstract; I believe some kind of explanation could be of use, although since I am not a radiologist, I cannot be sure about this.
Paragraph 2.5, what are LR+ and LR-? Specify that you mean likehood ratio, please.
Paragraph 3.1: the first lines are somewhat confusing. 111 patients were included from January 2018 to December 2019, you mean that 111 patients that received a diagnosis between January 2018 to December 2019 were included in the study? Also: It should be noted that two of them were excluded due to the small volume of their lesions. So the number is 109? Please clarify. Also: excluded due to the small volume of their lesions (3.0 and 1.4 mL) which is an impairment for the calculation of metabolic 18 F-FDG-PET heterogeneity due to the small number of voxels present in lesions < 10cc (16): this is a selection criteria, and should be stated in the methods, not in the results.
Discussion: "Moreover, smokers with lung cancer (mostly adenocarcinomas) have more tumour genetic mutations than non-smokers, according to the results of the sequencing-based studies [26–28]." What do you mean? I do not think you are referring to the targetable alterations in lung adenocarcinomas (i.e., EGFR, ROS1, ALK), do you mean the overall mutational burden? If so, what's the point of the sentence?
I also would suggest the Authors to add a radiological image highlighting their radiological "signature": that would be useful also for non-radiologists reading the work.
